# Advances in Plant Species Recognition Mediated by Root Exudates: A Review

**DOI:** 10.3390/plants14193076

**Published:** 2025-10-04

**Authors:** Fumin Meng, Renyan Duan, Hui Yang, Qian Dai, Yu Zhang, Jiaman Fu

**Affiliations:** 1College of Agriculture and Biotechnology, Hunan University of Humanities, Science and Technology, Loudi 417000, China; mengminfu@163.com (F.M.); daiqian0215@outlook.com (Q.D.); fujiaman@163.com (J.F.); 2Key Laboratory of Development, Utilization, Quality and Safety Control of Characteristic Agricultural Resources in Central Hunan Province, Loudi 417000, China

**Keywords:** root exudates, recognition effect, kin recognition, self-/non-self-recognition

## Abstract

Root exudates are critical signaling molecules in belowground plant–plant interactions, regulating physiological and ecological responses in adjacent plants through kinship recognition and self-/non-self-discrimination systems. This review systematically synthesizes the compositional diversity of root exudates, with particular emphasis on elucidating the ecological foundations of plant recognition modalities (kin recognition, allelopathy, plant self-/non-self-identification, and growth regulation). The analyses demonstrate that exudate composition is dynamically modulated by plant species identity, rhizosphere microbial communities, and environmental stressors, with signaling functions mediated through both physical signal transduction and chemical signal decoding. This chemical communication system not only drives species-specific interaction strategies but redefines the theoretical frameworks of plant community assembly by establishing causal linkages between molecular signaling events and ecological outcomes.

## 1. Introduction

Interspecific plant interactions constitute a fundamental aspect of ecological research, with the competition–cooperation paradox representing a key theoretical challenge. Under resource-limited conditions, plants demonstrate behavioral plasticity through either competitive resource acquisition or mutualistic coevolutionary associations [1,2,3]. These interactions fundamentally depend on neighbor recognition systems mediated by chemical signaling networks, primarily involving root exudates and volatile organic compounds, which facilitate spatial discrimination between conspecifics and heterospecifics to optimize resource allocation [3,4]. The recognition mechanism coordinates a hierarchical phenotypic response cascade comprising (1) individual-level morphological adaptations (e.g., shade avoidance) and reproductive strategy adjustments (e.g., propagule size modulation); (2) the physiological induction of defense systems (e.g., secondary metabolite biosynthesis); and (3) population-level niche specialization via fitness optimization [5,6]. This multiscale adaptive integration, ranging from molecular detection to ecosystem-scale patterns, provides a new paradigm for understanding plant community assembly. The emerging framework demonstrates how chemical signaling converts molecular recognition events into ecological consequences through coordinated phenotypic regulation.

Plant root systems serve as advanced ecological sensors that regulate interspecific recognition via adaptive signaling pathways. Specialized allelochemical communication networks allow roots to interpret complex soil environments, incorporating the detection of ionic gradients, hydraulic patterns, and moisture variations, while simultaneously enabling species-specific differentiation through the spatiotemporal distribution of VOCs and root-secreted metabolites [7,8]. A critical feature of this recognition system is its capacity for functional group discrimination: upon contact with heterospecific roots or different functional groups, plants initiate chemical demarcation through secreted signaling molecules, particularly strigolactones and flavonoids.

In contrast, conspecifics or functionally similar neighbors often establish resource-sharing networks [9,10,11]. *Pisum sativum* roots display significant spatial differentiation when exposed to *Festuca glauca* rhizosphere signals, characterized by altered primary root angles and redistributed lateral root biomass [6]. *Arabidopsis thaliana* (WS ecotype) shows ecotype-specific responses, increasing lateral root formation in soil conditioned by Col ecotypes while inhibiting lateral growth in conspecific exudate environments [12]. This recognition-dependent plasticity is particularly evident during competitive interactions: *Fagopyrum esculentum* and *Amaranthus retroflexus* exhibit reciprocal root avoidance, while conspecific interactions preserve vertical growth patterns [11]. These observations demonstrate an adaptive strategy where plants adjust root architecture through species recognition, optimizing the trade-off between resource acquisition and metabolic defense expenditure to enhance fitness in variable soil conditions.

The rhizosphere represents a dynamic plant–soil interface characterized by chemically defined gradient fields established through root exudation processes [7]. This root-influenced zone displays spatiotemporal heterogeneity in biogeochemical cycling patterns, primarily regulated by root metabolic activity [1,13]. In this microenvironment, roots release specialized metabolites that serve as phytochemical signals, forming complex interplant communication networks critical for species recognition. These signaling compounds mediate concentration-dependent ligand–receptor interactions, simultaneously coordinating both belowground root avoidance responses and aboveground developmental processes including flowering timing and stress adaptation.

Recent studies demonstrate that root exudation patterns exhibit marked spatiotemporal heterogeneity, regulated by three primary factors including host plant genetic factors, soil physicochemical properties, and rhizosphere microbial interactions. To elucidate the mechanisms of root secretion-mediated recognition, this review integrates current knowledge across four key aspects: (1) root exudate composition, (2) environmental and biological regulation of exudation dynamics, (3) plant–plant recognition systems, and (4) molecular mechanisms of secretion-mediated identification. This comprehensive analysis establishes a conceptual framework for understanding the complex rhizosphere communication network.

## 2. Method

A systematic search was conducted on Web of Science, Scopus, PubMed, SpringerLink, Wiley Online Library, and ScienceDirect. The following keyword strings were used: (“root metabolites” or “root exudates”) and (“self-/non-self-identification” or “kinship identification” or “identification effect”). Language was restricted to English, and document type was limited to original research articles and systematic reviews. Studies were included if they (i) focused on plants, (ii) reported an explicit experimental or observational design (e.g., randomized controlled experiments, cohort studies), or (iii) were systematic reviews or meta-analyses. Duplicate publications, conference abstracts, editorials, and studies lacking necessary data or results were excluded. Non-quantitative, conceptual diagrams were created in BioRender (https://biorender.com; Toronto, ON, Canada; version 2024.3) and exported as 600 dpi PNG files. Final vector editing was performed in Adobe Illustrator 2019 (v23.1.0, Adobe Inc., San Jose, CA, USA).

## 3. Composition of Root Exudates

Root exudates function as crucial chemical mediators in inter-root communication within plant–soil systems. These bioactive compounds consist of two main components: primary metabolites actively released by living root tissues and microbially modified secondary metabolites produced through rhizosphere processing [14,15]. Functionally, root exudates demonstrate dual regulatory roles in plant physiology: directly promoting plant growth through phytohormone-like activities and indirectly altering soil conditions via physicochemical changes. Notably, emerging evidence indicates these carbon-rich exudates form reciprocal interactions with rhizosphere microbiota, collectively enhancing host plant nutrient uptake efficiency and systemic resistance through microbiome-mediated metabolic adjustments [16,17].

Molecular analyses have systematically revealed root exudates as structurally hierarchical chemical mixtures. These secretions can be functionally divided into two groups: high-molecular-weight components such as mucopolysaccharides and defense proteins, and low-molecular-weight compounds forming complex metabolic networks. The low-molecular-weight fraction includes three main classes: primary metabolites (carbohydrates, amino acids, organic acids), specialized secondary metabolites (phenolics, terpenoids), and signaling molecules (flavonoids, strigolactones) [18,19]. Currently, more than 200 ecologically functional bioactive compounds have been structurally characterized, which can be classified into three categories based on their functions (Table 1).

Plant root exudates function as critical mediators in both kin recognition and allelopathic interactions, with organic acids, phenolics, flavonoids, terpenoids, and alkaloids constituting the core chemical signaling network [20,21]. For example, organic acids (e.g., oxalic acid) solubilize recalcitrant soil phosphorus, enabling preferential nutrient allocation to related plants through kin selection mechanisms [22,23]. Phenolic acids secreted by soybean roots in maize/soybean intercropping systems demonstrate antifungal activity against *Phytophthora sojae*, constituting a belowground defense strategy [24]. Flavonoids act as chemoattractants for beneficial microorganisms, simultaneously coordinating tripartite symbiosis involving rhizobia, arbuscular mycorrhizal fungi, and host plants [25].
plants-14-03076-t001_Table 1Table 1Composition category and ecological functions of root exudates.Functional ClassificationComposition CategorySecretionEcological FunctionNutrition and EnergySaccharideGlucose, fructose and galactose (monosaccharides), sucrose, lactose and maltose (disaccharides), raffinose and stachyose (oligosaccharides), glycogen, starch and cellulose (polysaccharides), etc.Bioenergy [26], store nutrients [27].ProteinGlobulin, glutelin, glycoprotein, lipoprotein, albumin [28], etc.Maintain biological growth and development, participate in metabolism, material transport [29].Organic acidCitric acid, tartaric acid, oxalic acid, malic acid, succinic acid [30,31], etc.Activation, reduction, pH adjustment [32].Amino acidGlycine, alanine [33], valine, leucine (non-polar amino acids), tryptophan, glutamic acid, tyrosine, serine, cysteine (polar amino acids) [30,34], etc.Improve the antioxidant rate and metabolic rate [30,35], an important component of the cell [36].Defense and Interaction CategoryPhenolic compoundsHydroxycinnamic acid, phenolic acid, phenolic lignan hydroxybenzoic acid, flavonol and condensed tannin [37], etc.Antioxidant [38], pest resistance, growth regulation [39].VolatilesIndole, hexanal [40], nonanal [41], etc.Promote plant–soil symbiosis and recruit beneficial bacteria [42].FlavonoidsQuercetin [43], naringenin, hesperidin, luteolin, isoprene flavonoids [44,45], etc.Antioxidant, anti-inflammatory, antibacterial, anti-tumor, immune regulation [46,47].Lactone5-Deoxystrigol [48], Strigol [49], etc.Promote hyphal branching and symbiosis of arbuscular mycorrhizal fungi [48]Regulation and Signal CategorySteroidStigmasterol, bile acid, fucosterol, rapeseed sterol, ergosterol [50], etc.Regulate drugs, glucose metabolism and energy metabolism, detoxification [51,52].Growth hormoneIndoleacetic acid, gibberellin, abscisic acid [53], etc.Rhizosphere signal substances, promote plant growth [54,55].EnzymeProtease, carbohydrate enzyme, lipase, amylase, pectinase [56], etc.Catalysis, maintain body function [57].Signal molecule hormoneEthylene, allantoin, jasmonic, etc.Growth regulators, important hormones for plant development and defense [58], and signal factors for root growth [59,60].

## 4. Factors Affecting Root Exudates

Root exudation regulation involves complex interactions among multiple factors, with classification systems varying by research focus [61]. Vives-Peris et al. [62] classified influencing factors into three groups: (1) chemical components (soil nutrients, phytohormones, trace elements), (2) physical parameters (light, temperature, CO_2_, humidity), and (3) biotic interactions (rhizosphere microbiota, fungi, soil fauna). Berg and Smalla [63] and Dessaux et al. [64] emphasized plant genetic determinants, particularly species- and genotype-specific exudation patterns. A later study by Lamichhane et al. [65] expanded these frameworks to include plant phenological stages, soil properties (pH, texture), and edaphic structures. Our study integrates these approaches, identifying three key regulatory categories: plant species/genotype variation, rhizosphere microbial dynamics, and abiotic stress.

### 4.1. Plant Species

Root exudate composition varies significantly across phylogenetic lineages, developmental stages, and genetic backgrounds. Drought-tolerant woody species (*Balanites aegyptiaca* and *Acacia raddiana*) exhibit distinct exudate profiles compared to drought-sensitive counterparts, reflecting evolutionary adaptations [66]. Developmental regulation is evident in model plants: *Arabidopsis thaliana* shifts from sugar-dominated exudates in seedlings to amino acid/phenolic compounds in mature stages, mirroring growth-to-defense transitions [67,68]. Genetic variation further drives exudate diversity, as demonstrated by differential organic acid secretion patterns in rice cultivars [69]. These findings highlight how phylogenetic, ontogenetic, and genetic factors collectively shape root exudate strategies.

### 4.2. Rhizosphere Microorganism

Root exudates act as key chemical mediators that shape rhizosphere microbiome assembly through soil modification [70,71,72]. Some evidence demonstrates bidirectional regulation: pathogens like *Pseudomonas syringae* pv. tomato DC3000 induce amino acid and jasmonic acid secretion while reducing carbohydrates in *Arabidopsis thaliana*, mediated by jasmonic acid signaling for beneficial microbiota recruitment [73]. Similarly, *Bacillus* spp. colonization in *Solanum lycopersicum* triggers glycosylated azelaic acid derivatives as ecological modulators [74]. This microbe–plant metabolic dialog exemplifies co-evolutionary mechanisms, balancing resource allocation between growth and defense through exudate-mediated feedback loops.

Microbial communities play a dual role as regulators and amplifiers in ecosystems through secretion mediated signaling, rather than simply filters. Their mechanisms of action are highly environment-dependent and species-specific [2,3,75]. For example, microorganisms selectively respond to plant signals by secreting specific metabolites such as flavonoids, phenolic acids, and quorum-sensing molecules, forming a bidirectional regulatory network [76,77]. The hyphae of arbuscular mycorrhizal fungi connect different plant roots, converting the carbon sources (such as sugars) secreted by plants into a “highway” for cross-species signal transmission, expanding the communication range between plants [78]. In addition, microbial communities exhibit selectivity in the degradation or transformation of plant secretions, rather than passive filtration [76].

### 4.3. Abiotic Stress

Abiotic stress has been demonstrated to exert a substantial influence on the quantity and composition of root exudates [79,80,81]. Drought stress has been demonstrated to induce alterations in cell membrane permeability, thereby affecting the passive secretion of small molecule organic compounds, such as sugars and organic acids, whilst concomitantly increasing the secretion of defensive secondary metabolites, including flavonoids and phenolic acids [82,83,84]. This process has been shown to facilitate the recruitment of drought-tolerant microorganisms, such as arbuscular mycorrhizal fungi. In conditions of nutrient stress, plants have been observed to activate soil insoluble nutrients by increasing the secretion of organic acids, such as citric acid and malic acid, while concurrently secreting more flavonoids to recruit nitrogen fixing and phosphorus solubilizing bacteria [18]. In the context of salt stress, plants have been observed to upregulate the secretion of compatible solutes, such as proline and betaine, to ensure the maintenance of cellular osmotic balance [85,86]. Concurrently, climate fluctuations have been observed to impact plant signal transduction through the alteration of rhizosphere microbial diversity and the consequent effect on rhizosphere metabolite secretion [87]. For example, the decline in pathogenic microorganisms triggered by climate change has the potential to diminish the signals that facilitate plant communication [88]. Furthermore, it has been demonstrated that extreme droughts, exacerbated by climate change, may result in an increased secretion of allelopathic substances (e.g., phenolic acids) by plants, thereby hindering the growth of neighboring species that are sensitive to such substances and, consequently, altering community structure [89].

## 5. The Mode of Action of Root Exudates Mediated Plant Recognition

### 5.1. Define Kinship

The ecological relevance of plant kin recognition systems continues to generate scientific controversy [90]. Numerous studies contest the kin recognition hypothesis, suggesting instead that genetic relatedness rather than active discrimination primarily determines plant phenotypic plasticity. Milla et al. [91] performed extensive fitness assessments, showing that the kin selection model insufficiently explains phenotypic variation patterns across plant evolutionary trajectories. Corroborating this view, Nord et al. [92] conducted controlled experiments with *Phaseolus vulgaris* (kidney bean), finding that root distribution patterns were mainly determined by soil resource distribution rather than recognition processes. These results collectively indicate that environmental conditions and genetic associations likely outweigh active recognition in mediating plant–plant interactions.

Experimental studies increasingly reveal that plants utilize complex chemical communication networks to regulate interspecific relationships [93]. *Triticum aestivum* actively inhibits adjacent plant growth via allelochemical production [60,94], while *Oryza sativa* displays genotype-specific root architecture modifications when growing with related conspecifics [95,96]. These observations imply that plants evolved recognition systems functionally comparable to animal kin selection. Importantly, plant–microbe symbioses chemically mediate these interactions, as demonstrated by arbuscular mycorrhizal flavonoids that modify host root development, suggesting underground recognition systems may promote niche partitioning via plant–microbe–plant signaling cascades.

Recent studies have established two primary underground signaling mechanisms mediate plant kin recognition: common mycorrhizal networks (CMNs) and root exudate signaling. CMNs, formed by interconnected fungal hyphae, establish symbiotic interfaces for bidirectional resource transfer, with host plants allocating photosynthetic carbon to fungal symbionts in exchange for improved mineral uptake and stress resistance [97]. These networks function as kin recognition systems, as shown on *Pseudotsuga menziesii* demonstrating kin-preferential carbon transfer via CMNs [98]. Such discriminatory allocation implies fungal-mediated signal transmission, though specific signaling molecules remain uncharacterized. Simultaneously, rhizosphere fungi alter soil microstructure to enhance root exudate and volatile organic compound diffusion [2,99,100]. Arbuscular mycorrhizal fungi (AMF) can modulate rhizosphere metabolites (e.g., lipochitooligosaccharides, secreted proteins, salicylic acid, and jasmonic acid) to regulate inter-species partner recognition, colonization, and symbiotic relationships [2,101]. This metabolic signaling cascade subsequently activates plant defense mechanisms, enhancing host tolerance to biotic and abiotic stresses. For example, pathogen-infected tomatoes transmit early-warning signals through mycorrhizal networks to neighboring plants, inducing systemic resistance [102]. The cell wall-degrading enzymes (e.g., cellulases), secreted by *Rhizoctonia* spp. and *Fusarium* spp., can be recognized by plant pattern recognition receptors, thereby activating pattern-triggered immunity [103,104]. Key unresolved questions include the following: CMN transmembrane signaling mechanisms, spatiotemporal signal distribution patterns, and energy metabolism supporting hyphal connections. Preliminary data suggest membrane vesicle transport may mediate intercellular signaling, but molecular components and metabolic pathways need further characterization [105].

Root exudates serve as a sophisticated signaling system for kin discrimination in plants, with phylogenetically related individuals exhibiting conserved metabolic profiles [15,106]. Strigolactones display kin-specificity in arbuscular mycorrhizal fungal colonization, optimizing nutrient exchange efficiency between related individuals [107]. Phenolic acids (e.g., cinnamic acid derivatives) demonstrate dose-dependent allelopathic effects, with stronger inhibition observed against non-kin competitors [108]. At the level of signal transduction, plants have evolved sophisticated receptor systems to recognize these chemical signals. The receptor like kinase family proteins, such as FERONIA, can specifically sense peptide signaling molecules secreted by the root system [109]. The synergistic effect of these receptor systems enables plants to accurately distinguish between self, related, and non-related individuals and make corresponding physiological responses.

Crop kin recognition constitutes an evolutionary adaptation that optimizes resource distribution and improves population fitness, with *Oryza sativa* serving as a principal model. This species utilizes root exudates as chemical cues for kin discrimination, effectively suppressing weeds while reducing allelopathic expenditure, thus enhancing growth compatibility among relatives [110]. While root exudates are confirmed mediators of kin recognition, their mechanistic basis requires further investigation [111]. Research indicates three potential recognition compound classes: secondary metabolites (e.g., rice allantoin) [112,113], phytohormones (e.g., strigolactones, jasmonates) [114,115], and specialized signals (e.g., ryegrass lactone) [116]. These chemical mediators facilitate complex resource allocation decisions during competitive interactions. Future studies should focus on CMN–root exudate crosstalk to decipher integrated underground communication systems.

### 5.2. Allelopathy

Root exudates function as complex chemical mediators that interpret phylogenetic signals and facilitate plant–plant communication, coordinating ecological adaptation mechanisms (Figure 1). These secretions comprise various allelochemicals, notably flavonoids and terpenes, displaying concentration-dependent regulatory properties: (1) signaling pathways—water-soluble molecules mediate short-distance signaling while volatile organic compounds participate in long-range communication; (2) biological effects—stimulatory interactions at low concentrations versus inhibitory effects at elevated concentrations; and (3) ecological impacts—rhizosphere microbiome modification, neighboring plant behavior regulation, and community-wide response synchronization. This chemical signaling network constitutes an evolutionary strategy that enhances plant fitness through accurate environmental perception and coordinated responses [117].

Allelopathy, first systematically described in the 1920s, constitutes an advanced biochemical signaling network where plant- and microorganism-derived secondary metabolites mediate dual-directional (either promotive or suppressive) interactions with adjacent organisms via specific molecular mechanisms [118]. Contemporary studies have identified four primary allelopathic transfer pathways: root exudation, aerial organ volatilization, plant residue breakdown, and precipitation-driven leaching. Among these, rhizospheric communication through root exudates has been established as the predominant controller of vegetation structure, interspecies coexistence relationships, and ecosystem succession processes [4]. These allelochemical systems coordinate intricate spatial-temporal competition dynamics through three key mechanisms: metabolic signaling cascades, dose-dependent physiological responses, and taxon-discriminative recognition pathways (Figure 1).

Allelochemicals function as essential mediators in plant–plant interactions, significantly influencing neighboring plants’ physiological and ecological characteristics through specific chemical interference mechanisms, particularly affecting root system development and hypocotyl growth [119]. Current research has substantially enhanced our comprehension of invasive species’ allelopathic tactics, uncovering three primary mechanistic aspects: (1) Comprehensive invasion approaches—*Schinus terebinthifolius* utilizes root exudates that modify rhizosphere microbial communities, selectively increase arbuscular mycorrhizal fungi abundance, strengthen nutrient acquisition efficiency, and elevate stress resistance [120]. (2) Biochemical interference processes—*Eupatorium adenophorum* releases phytotoxic compounds that disturb mineral absorption mechanisms, interfere with amino acid production, modify secondary metabolism patterns, and particularly block phosphate uptake routes [121]. (3) Ecosystem-level impacts—these chemical interactions generate competitive imbalances, promote ecological niche establishment, and influence vegetation structure evolution. These discoveries collectively demonstrate allelopathy’s role as a crucial determinant of plant competitive relationships via complex biochemical communication systems.

Recent breakthroughs in allelopathy research have elucidated the molecular recognition and ecological functions underlying plant chemical communication. Plants secrete secondary metabolites (e.g., phenolic acids, terpenoids) to mediate interspecies information transfer through two primary pathways: volatile organic compounds (VOCs, such as monoterpenes) that activate defense systems in neighboring plants via atmospheric diffusion, and water-soluble compounds (e.g., coumaric acid, ferulic acid) that reshape rhizosphere microbiomes through root exudation or rain leaching [4,122]. This chemical signal has a dual ecological effect, which can suppress competing species while promoting mutualistic symbiosis. Notably, plant responses to chemical cues demonstrate high specificity, with effect intensities dynamically modulated by evolutionary kinship, environmental parameters (light, temperature, moisture), and microbial communities [4]. Such complexity underscores the evolutionary significance of allelopathy as a biological alarm system, enabling stress-coordinated defense at the community level through chemical early-warning signals.

Some advances have elucidated the intricate regulatory networks underlying plant-specific recognition of allelochemicals. Xie et al. [122] identified receptor protein RBP47B, which binds and senses diverse phenolic acid allelochemicals, including salicylic acid, protocatechuic acid, coumaric acid, and ferulic acid. This receptor forms stress granules via liquid–liquid phase separation, thereby inhibiting target plant translation processes and mediating interspecific competition. This “chemical receptor–signal transduction–phenotype output” cascade provides a molecular framework for understanding allelopathic inhibition. Building upon this, the allelobiosis theory proposed by Kong et al. [4] reveals that plants can chemically fingerprint neighbor identity (e.g., kin vs. non-kin) and dynamically adjust resource allocation strategies. Such “chemical recognition–resource optimization” synergy is particularly evident in mixed cropping systems [4]. For instance, potato-derived paclitaxel reshapes tomato rhizosphere microbiomes, selectively activating *Verticillium wilt* resistance genes [123], while terpenes from red pine trigger ginseng MAPK signaling pathways, significantly reducing *Alternaria infection* rates [124]. These findings collectively establish a cross-species communication model of chemical signal-receptor interaction-defense enhancement. However, current research predominantly focuses on allelopathic suppression mechanisms, while the molecular pathways of facilitative interactions (e.g., microbiota-mediated allelopathic facilitation) remain underexplored [125], representing a critical frontier for future investigation.

Autotoxicity, a distinct type of intraspecific chemical interaction, functions as an ecologically important self-inhibition system mediated by plant-produced secondary metabolites that limit population regeneration processes [126]. This autotoxic effect has been recognized as a major contributor to agricultural replant problems. Specifically, the long-term cultivation of *Panax notoginseng* produces root exudate-modified soil environments that markedly reduce cell division activity in seedling root apical meristems [127]. Correspondingly, the extended monoculture of potato (*Solanum tuberosum*) alters the rhizosphere microbial balance, generating growth-limiting conditions [128]. Together, these results confirm autotoxicity as a concentration-sensitive ecological control mechanism mediated by complex biochemical signaling networks.

### 5.3. Plant Self-/Non-Self Identification

Plant self-/non-self-recognition constitutes an advanced biological mechanism primarily regulated by molecular identification processes during root interactions [129]. Plants display significant phenotypic flexibility via dynamic root exudation patterns, functioning as an advanced chemical signaling system for interspecies communication [4]. Root-derived specialized metabolites, including terpenoids, phenolic acids, and nitrogen containing compounds, serve dual biological roles as both stress-responsive molecular markers and constituents of intricate allelochemical networks involved in environmental perception [130]. The facultative parasitic species *Triphysaria versicolor* exemplifies this phenomenon, precisely regulating haustorium-inducing factors in its root exudates to discriminate among potential hosts during parasitic interactions [131]. These findings uncover evolutionarily maintained niche-partitioning strategies governed by metabolic adjustments, providing fundamental insights into the chemical ecology of plant–plant interactions.

Root exudates function as complex chemical mediators of plant phylogenetic recognition through taxon-specific response systems, promoting ecological niche differentiation among related species (Figure 1). This mechanism is clearly demonstrated in *Ambrosia artemisiifolia*, where root secretions trigger differential avoidance responses in three key crops (*Triticum aestivum*, *Glycine max*, and *Zea mays*), effectively minimizing interspecies competition via spatial separation [132]. Niche differentiation serves as a fundamental ecological principle for analyzing plant interspecific relationships, though investigations in this field encounter distinct methodological constraints relative to animal systems. The immobile growth habit of plants, combined with their fixed architecture and delayed phenotypic adjustments, presents substantial challenges in deciphering both intraspecific and interspecific interaction mechanisms. Chemical-mediated neighbor recognition in plants displays two principal features: significant temporal accumulation effects in behavioral responses and considerable species-specific variation in response patterns. Experimental data confirm that environmental heterogeneity acts as a key determinant of adaptive root allocation strategies, which are finely tuned based on neighboring plant identity. For example, *Plantago lanceolata* demonstrates substantially enhanced root biomass allocation when grown with *Centaurea cyanus* in uniform soil conditions, while showing preferential root growth toward *Poa annua* neighbors in heterogeneous resource environments [10]. These results not only clarify the plasticity mechanisms regulating plant–plant interactions but also confirm the presence of advanced recognition systems enabling species-specific response adjustments.

Plant self-/non-self-recognition systems display exceptional biological specificity operating across multiple organizational scales. In reproductive systems, foundational research by Nasrallah [133] established that self-incompatible *Arabidopsis thaliana* ecotypes can reliably distinguish self- from non-self-pollen, thereby controlling pollination outcomes, offering critical evidence for interplant recognition processes. Modern plant immunity studies have further clarified that cell surface-localized pattern recognition receptors facilitate self-/non-self-discrimination through pathogen-associated molecular pattern detection, initiating pattern-triggered immunity with precisely coordinated defense responses [134]. Importantly, these recognition processes operate beyond reproductive contexts. Research on clonal species demonstrates that vegetatively propagating *Ipomoea nil* establishes synchronized defense systems among ramets via belowground signaling, revealing whole-plant integration through self-recognition [135]. Systematic analyses of shoot–root interactions confirm that rhizosphere signaling networks and resource competition significantly shape aboveground phenotypic responses, with defense coordination being governed by root-mediated self-identification mechanisms.

### 5.4. Adjust the Growth Strategy

As Rasmann et al. [136] demonstrate, root-secreted compounds function as pivotal regulators, coordinating plant growth patterns, shaping rhizosphere conditions, and facilitating intricate biological interactions. Within plant communities exhibiting niche overlap, the dynamic relationship between spatially variable soil resources and shared ecological requirements promotes evolutionary adjustments via three interrelated plastic responses: refined resource competition approaches, belowground niche differentiation, and the induction of stress preparedness systems (Figure 1). These coordinated adaptations enable coexisting plants to successfully manage the constraints of shared habitat utilization [137].

It has been established that root exudates are responsible for orchestrating root architecture remodeling through both hormone-dependent and hormone-independent pathways. It has been demonstrated that rhizospheric microorganisms (e.g., arbuscular mycorrhizal fungi, rhizobia) have the capacity to secrete phytohormones, including auxin and cytokines, which in turn activate the hormonal signaling network that governs lateral root development. This process is mediated by key transcriptional regulators such as auxin response factors and auxin response regulators [82,138]. Pattern recognition receptors (PRRs), including flagellin-sensitive 2 (FLS2) and Brassinosteroid insensitive 1-associated kinase 1 (BAK1), directly perceive microbe-associated molecular patterns (MAMPs). This, in turn, triggers mitogen-activated protein kinase (MAPK) cascades, leading to immediate root architectural adjustments [138].

Moreover, root exudates can initiate molecular mechanisms of defense through both direct and indirect defense pathways. The mixture of amino acids and long-chain organic acids secreted by plants after pathogen infection can directly inhibit the growth of pathogens, and their receptors are lipid transporters on the pathogen cell membrane [139]. Secretions indirectly inhibit pathogens by enriching beneficial bacteria such as *Pseudomonas*, and their signal transduction involves the recognition of microbial metabolites by plant NOD like receptors [82].

## 6. Plant Recognition Mechanism Mediated by Root Exudates

Plants demonstrate three characteristic root architectural responses: proliferation, avoidance, and neutral. When encountering neighboring roots in nutrient-heterogeneous soils [140], root exudates mediate these interactions as primary rhizospheric signaling agents. The recognition mechanisms function through dual pathways: (1) physical regulation involving oscillatory patterns that synchronize internal signaling with external detection, and (2) chemical signaling via specialized metabolites acting as molecular cues [141].

### 6.1. Physical Signal Regulation Mechanism

Oscillatory signals arising from plant electrophysiology or endogenous hormone dynamics may function as essential mediators of mutual recognition, potentially serving as physiological communication signals between roots. The oscillation hypothesis proposes a physiologically coordinated self-/non-self-discrimination system in root systems. Within this framework, genetic identity recognition among clonal ramets depends mainly on physiological synchronization via interconnected root networks, displaying strict spatial constraints. Notably, physically separated ramets shift from self-recognition to non-self identification, implying hormone-regulated oscillatory mechanisms in root recognition [5,142].

The oscillation hypothesis suggests resonance frequency serves as a key parameter for plant root self-/non-self-discrimination [5]. *Ambrosia artemisiifolia* demonstrates this principle through kinship-dependent root recognition: frequency matching between physiologically identical roots reduces growth rates while promoting cooperative resource allocation, whereas frequency mismatch with distinct roots triggers competitive growth acceleration [143]. These mechanisms likely stem from heritable oscillatory responses [144]. *Fragaria* species show physical connections between ramets enhance clonal performance via root segregation, suggesting physiologically integrated oscillations as the mechanism [144]. Phytohormones regulate root integration through internal synchronous oscillation mechanisms, as demonstrated by auxin-mediated oscillatory gene expression during *Arabidopsis* root branching [141]. Behavioral evidence supports this model: *Cayratia japonica* tendrils favor coiling around disconnected self-ramets, while roots typically approach non-self-competitors but avoid clonal relatives [142,145]. These findings indicate endogenously synchronized oscillations combined with external signals create a biological framework for environmental perception and self-recognition [146].

Root self-/non-self-recognition exhibits ecological importance through physiologically regulated behavioral plasticity that optimizes either cooperative resource integration or competitive acquisition (Figure 2). The underlying mechanism involves root-emitted identification signals encoded as oscillatory frequencies via internal synchronous oscillation (ISO) processes. These frequency-dependent signatures function dually: as intracellular recognition triggers activated by frequency detection and as extracellular signals released into the rhizosphere. Self-recognition emerges when nodal connections enable frequency matching, generating resonance amplification that forms the physiological foundation of kin discrimination. In contrast, disconnected ramets displaying frequency mismatches cannot establish ISO-mediated communication, resulting in non-self-identification. This dichotomous recognition system ultimately determines whether roots adopt resource-sharing or competitive allocation strategies.

### 6.2. Chemical Signal Regulation Mechanism

Plant–plant interactions are regulated through integrated signaling networks involving both physical (electrical signals) and chemical (secondary metabolites, nutrient gradients) communication pathways. These signaling systems trigger various allelopathic responses such as shade avoidance, root foraging plasticity, and chemical defense activation (Figure 3). Root exudates containing allelochemicals play a pivotal role in plant recognition processes. Competitive interactions between neighboring plants, as established by Bais et al. [147], emerge through three principal mechanisms, namely resource competition, chemical interference, and allelopathic inhibition in parasitic relationships. This leads to a key ecological consideration: whether allelopathic traits provide growth benefits under resource-limited conditions. Invasive species studies demonstrate this phenomenon clearly, the allelopathic invader *Centaurea diffusa* shows markedly superior competitive performance against native North *American Centaurea* species (non-allelopathic) during invasion [148], confirming the evolutionary advantage of allelopathic traits in interspecific competition.

Recent studies have systematically elucidated allelochemical-mediated recognition mechanisms in rhizosphere environments (Figure 3). Major allelopathic compounds including jasmonates and benzoxazolinones act as molecular signals that selectively activate defense-related phytohormone pathways (JA/SA) in neighboring plants, thereby altering their resource allocation strategies. These observations support the tripartite interaction model of Bais et al. [147], where plant–plant competition results from combined effects of resource limitation, allelochemical inhibition, and parasitic interactions. Importantly, allelopathic effects persist beyond active secretion periods. Kong et al. [4] demonstrated that residual allelochemicals (phenolic acids, alkaloids) maintain activity through decomposition processes, causing sustained metabolic interference. A notable example is ferulic acid-induced phytotoxicity during crop residue decomposition, which suppresses root growth and aggravates replant disease symptoms [149]. These post-secretion effects highlight the ecological longevity of chemical recognition signals in agricultural ecosystems.

Some studies have comprehensively characterized multiple root-secreted signaling molecules that facilitate direct plant–plant communication via root exudates (Table 2). Key bioactive compounds such as (-)-loliolide, jasmonic acid (JA), ethylene (ETH), strigolactones (SLs), and allantoin have been functionally validated as essential mediators of interspecific interactions [140,150]. This section briefly summarizes these chemical signals and their specific functions within rhizosphere signaling networks.

The carotenoid-derived metabolite (-)-loliolide has been identified as a crucial root-secreted signaling compound involved in plant developmental regulation and interspecies communication [158]. Research has established its diverse functions in controlling flowering time, growth patterns, and defense mechanisms [159,160]. As a rhizosphere signal, (-)-loliolide facilitates neighbor detection and adaptive responses [161]. A representative example was documented in *Arabidopsis thaliana*-*Nicotiana benthamiana* interactions, where Arabidopsis-derived (-)-loliolide upregulated nicotine biosynthesis genes (*QPT*, *PMT1*) in tobacco roots, an effect eliminated by the genetic disruption of (-)-loliolide production [152]. These results definitively confirm (-)-loliolide as an authentic plant–plant signaling molecule. Remarkably, (-)-loliolide-mediated competition occurs independently of direct root contact or mycorrhizal networks. Studies showed allelopathic wheat varieties produce defensive benzoxazines in response to (-)-loliolide, verifying its function as a universal belowground competition signal. Together, these findings demonstrate (-)-loliolide’s essential role in plant neighbor recognition and resource competition [116].

Jasmonic acid (JA), an essential phytohormone ubiquitous in higher plants, critically regulates developmental processes and stress adaptation mechanisms [162]. The biosynthetic pathway originates from α-linolenic acid conversion to 13S-hydroperoxynicotinic acid (13-HPOT), followed by its transformation into 12-oxo-phytodienoic acid (OPDA) through sequential enzymatic reactions mediated by allene oxide synthase (AOS) and allene oxide cyclase (AOC) [77]. The subsequent peroxisomal oxidation of OPDA generates (+)-7-iso-JA, which undergoes ATP-dependent conjugation with isoleucine (Ile) catalyzed by jasmonate-amido synthetase (JAR1), yielding the bioactive jasmonoyl-isoleucine (JA-Ile) conjugate [163]. This principal bioactive form orchestrates multifaceted physiological responses via the JA signaling network.

Contemporary research has definitively characterized jasmonic acid (JA) as a master regulator coordinating plant defense systems against various biotic and abiotic challenges [161,164,165]. This hormone-dependent protective mechanism is particularly evident in pathosystem investigations using *Fusarium oxysporum*-infected tomato plants (*Solanum lycopersicum* cv. Momotaro), where magnesium oxide-triggered oxidative stress was demonstrated to amplify JA-mediated signaling pathways, consequently improving disease resistance [166]. These observations confirm JA’s pivotal role as a molecular activator initiating plant immunity following pathogen detection.

Jasmonic acid (JA) exhibits multifaceted defensive functions that encompass not only pathogen resistance but also herbivory protection, as demonstrated by its capacity to improve aphid resistance in wheat (*Triticum aestivum* L.) [167]. In addition to its intracellular signaling properties, JA and its volatile derivative methyl jasmonate (MeJA) have been identified as crucial components in plant–plant communication networks [163]. This signaling phenomenon has been extensively characterized in rice (*Oryza sativa*), where MeJA perception activates neighboring plant recognition mechanisms and induces adaptive morphological changes [168]. Supporting evidence from wheat studies (cv. Jing411) solidifies JA’s position as an interspecies signaling molecule, thereby elucidating its dual role in both defensive responses and interplant communication [116].

Ethylene, a gaseous plant hormone, functions as a principal coordinator of physiological and morphological adaptations to both environmental stimuli and developmental signals [169,170]. While its aerial roles are well-documented, ethylene has gained recognition as a crucial signaling molecule in rhizosphere interactions, mediating belowground plant communication [171,172]. This interplant signaling mechanism is particularly evident in peanut-cassava (*Arachis hypogaea*-*Manihot esculenta*) intercropping systems, where cyanide released by cassava stimulates ethylene production in peanut roots, facilitating interspecies recognition [173]. Notably, this ethylene-dependent signaling pathway orchestrates rhizosphere microbiome composition, leading to improved peanut productivity and stress tolerance via microbial-mediated processes [173].

Strigolactones (SLs), a group of root-synthesized carotenoid-derived plant hormones, have been increasingly recognized as pivotal regulators governing both developmental processes and ecological relationships [174,175,176]. These versatile metabolites operate through dual mechanisms, functioning simultaneously as internal growth modulators and external signaling agents. Within plant systems, SLs regulate shoot morphology and enhance arbuscular mycorrhizal symbiosis through the induction of fungal hyphal proliferation [177,178,179]. Externally, they act as essential signaling molecules in interplant communication, mediating neighbor identification processes. This function is particularly well-demonstrated in wild-type pea (*Pisum sativum*), where root-exuded SLs enable early recognition of neighboring plants [115].

Allantoin, a nitrogen-containing compound biosynthesized through the purine catabolic pathway, exhibits widespread distribution among diverse plant species [157,180]. Current research reveals allantoin’s significant involvement in hormonal cross-talk, specifically modulating abscisic acid and jasmonic acid signaling cascades [181,182]. In addition to its metabolic roles, allantoin has gained prominence as a key extracellular signaling agent in interplant communication [140,183]. This function is particularly evident in rice, where root-secreted allantoin serves as a rhizospheric signal for neighbor detection [184]. Most remarkably, contemporary investigations have established that allantoin-dependent signaling pathways regulate carbon allocation and reproductive strategies in rice, consequently shaping its species recognition characteristics [112].

Although root-derived chemical signals are universally present in the rhizosphere, substantial uncertainties remain concerning their exact molecular mechanisms in mediating interspecific recognition among plants. While contemporary studies have partially deciphered belowground interactions involving specific signaling molecules (e.g., (-)-loliolide and methyl jasmonate), three fundamental questions require further clarification: the spatial and temporal patterns of signal exudation, especially concentration-dependent response thresholds; the structural determinants of interspecies recognition specificity, particularly ligand–receptor interaction dynamics; and the complete spectrum of physiological response pathways induced by these signals, including the chronological regulation of defense-associated gene expression. These knowledge gaps underscore the necessity for comprehensive studies to develop an integrated framework of rhizosphere signaling systems.

## 7. Conclusions and Foresight

This review synthesizes contemporary knowledge of plant root exudates, with particular emphasis on their chemical diversity, biological functions, and participation in self-/non-self-discrimination and kin recognition mechanisms. Notwithstanding the persistent methodological variations in experimental approaches, especially with regard to the selection of plant material, genetic relatedness assessment, and the standardization of growth conditions, mounting experimental data consistently demonstrates the critical role of root-derived metabolites in recognition phenomena spanning diverse plant taxa.

Despite significant advancements in rhizosphere research, root-mediated plant identity recognition remains an emerging field with critical knowledge gaps at both fundamental and applied levels [2,3]. Current investigations represent only the initial phase of this complex research domain, where methodological challenges in deciphering plant communication mechanisms under natural soil conditions persist. Future interdisciplinary research should adopt a multi-scale approach integrating root exudate signaling across molecular, physiological, and ecological dimensions. At the molecular level, emphasis should be placed on elucidating chemotactic receptor-mediated signal transduction pathways and their subsequent modulation of defense gene expression. Physiological investigations should focus on unraveling autoregulatory mechanisms governing nutrient acquisition and stress responses, as well as exudate-mediated microbial recruitment strategies. Ecologically, research priorities include elucidating the contributions of these processes to biogeochemical cycling and interspecies recognition through metabolite–microbe–pathogen tripartite interactions. This integrative framework will not only enhance our fundamental understanding of plant communication networks but also facilitate translational applications in sustainable agriculture through targeted manipulation of rhizosphere signaling pathways.

To achieve practical agricultural applications, three synergistic strategies should be proposed. Firstly, exudate-targeted breeding can be used to select genotypes with optimized metabolite profiles for the suppression of pathogens and the recruitment of beneficial microbes. Secondly, metabolic engineering should be employed to precisely modulate exudate composition through pathway modification. Thirdly, microbiome management should be used to design microbial assemblages that synergize with host exudates for enhanced nutrient mobilization and stress tolerance. The integration of these strategies with precision agricultural technologies holds substantial promise for developing scalable solutions to sustainably intensify crop production systems.

## Figures and Tables

**Figure 1 plants-14-03076-f001:**
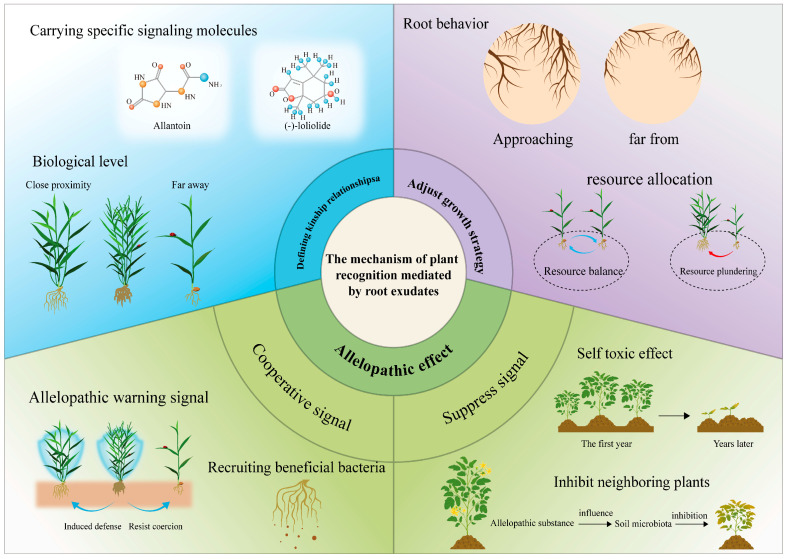
Overview of the role of root exudates in mediating plant recognition.

**Figure 2 plants-14-03076-f002:**
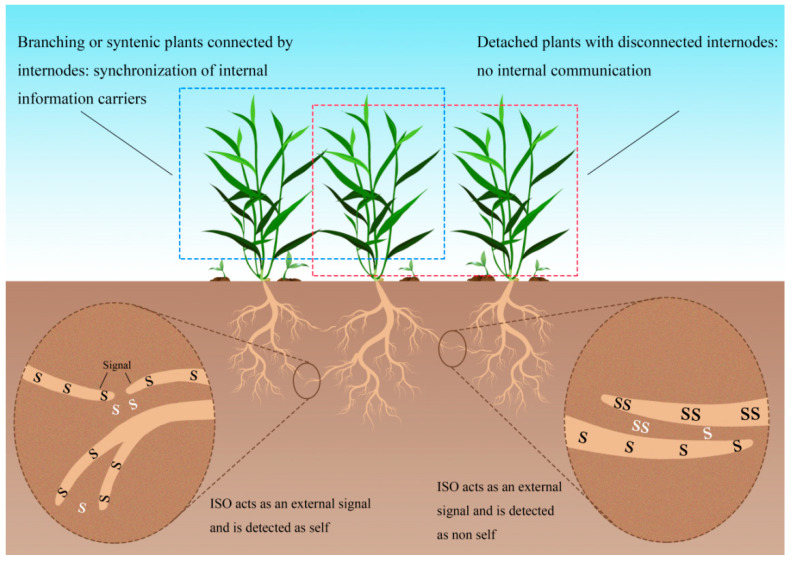
The possible mechanism of self-/non-self-recognition by oscillation hypothesis, internal synchronization and external identification. S and SS show different oscillation modes. Black S and SS showed root internal oscillation frequency, while white S and SS showed external communication (Adapted from Chen et al. [141]).

**Figure 3 plants-14-03076-f003:**
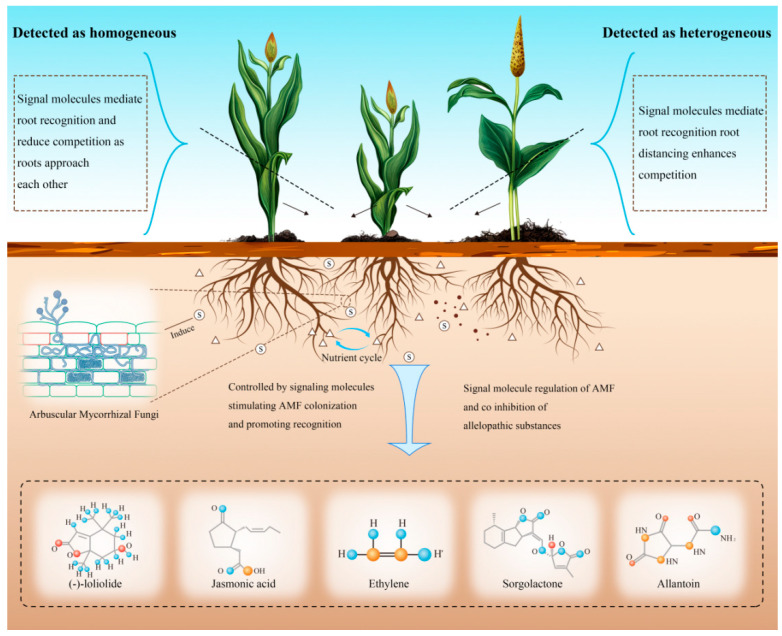
The chemical signal mechanism of underground root secretion, through several possible root secretion signal transduction chemicals to make the target plant recognize adjacent plants and induce a series of growth strategies. The small triangle in the figure represents nutrients, and S represents chemical signaling substances.

**Table 2 plants-14-03076-t002:** Important root secretory signal transduction chemicals and their characteristics among plants.

Signal Transduction Chemicals	Chemical Formula	Category	Characteristic
jasmonic	C_12_H_18_O_3_	oxidized monocarboxylic acid	Growth regulators, important hormones for plant development and defense [58,151], and signal factors for root growth [60].
(-)-loliolide	C_11_H_16_O_3_	ester compounds	The signal factors of root recognition are involved in the recognition and regulation of plant underground defense and induce the allelopathy of adjacent plants [138,152,153].
ethylene	C_2_H_4_	phytohormone	It regulates plant growth and development, the core regulator of root growth, underground root conduction signals [59], and participates in plant underground identification [154].
strigolactone	C_17_H_14_O_5_	sesquiterpene plant hormones of carotenoids	Root-derived chemical signals are involved in the regulation of rhizosphere signals [155], regulating plant development processes and responding to environmental changes [156].
allantoin	C_4_H_6_N_4_O_3_	imidazole heterocyclic compounds	Nitrogen-rich compounds, plant growth regulators [157], defense plants by biological stress [140].

## Data Availability

No new data were created or analyzed in this study.

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
