# Peer review of "Advances in Plant Species Recognition Mediated by Root Exudates: A Review"

_plants, 2025, doi:10.3390/plants14193076_

Round 1
Reviewer 1 Report
Comments and Suggestions for Authors
The review is well-structured, progressing logically from basic concepts to influencing factors, recognition mechanisms, and future directions. It effectively synthesizes diverse studies across model and crop species, highlighting practical implications for agriculture, such as allelopathy in invasive species control. The inclusion of tables (e.g., Table 1 on exudate classification and functions, Table 2 on signaling chemicals) and figures (e.g., Figure 1 overview, Figure 2 oscillation hypothesis, Figure 3 chemical signaling) enhances clarity and provides useful summaries. References are up-to-date (many from 2020-2024) and span interdisciplinary fields, demonstrating thorough literature coverage. The discussion of controversies, such as debates on kin recognition mechanisms, adds balance and depth. The manuscript has the potential to be a valuable contribution to the literature after addressing the points outlined below.
In Table 1, categories like "A high molecular" and "A low molecular" seem incomplete or typographical errors (e.g., should be "High-molecular-weight" and "Low-molecular-weight").
Table 2's references are aggregated for some entries, making it hard to trace specific claims.
Reviewer 2 Report
Comments and Suggestions for Authors
This review of root exudates in belowground plant interactions presents an interesting and well-executed collection of information from scientific studies on the regulation of physiological and ecological responses in adjacent plants through kin recognition and self/non-self discrimination systems. This work is worthy of scientific interest and promotes further study in physiology, microbial ecology, biological control, and molecular biology/genetics.
Line 56: Pisum sativum in italics
Line 57: Festuca glauca in italics
Line 59: Arabidopsis thaliana
All Latin scientific names in italics
Lines 68 and 78: The same concept is repeated; it is possible to summarize the two paragraphs in one.
Line 395: Include bibliographic references for the latest scientific research dedicated to identifying allelopathic relationships.
Table 1: It would be nice to be able to better lay out the bibliography referring to enzymes and amino acids
It would be nice to include cases of fungal interaction and report filamentous fungi present in the rhizospora to increase the value of the review (e.g. Rhizoctonia, Fusarium, Alternaria..)
Reviewer 3 Report
Comments and Suggestions for Authors
The manuscript entitled “Advances in Plant Species Recognition Mediated by Root Exudates: A Review” presents a systematic review on the chemical diversity of root exudates and their functions in regulating root systems, defense mechanisms, and phenotypic plasticity in relation to plant-plant and plant-microbe interactions within shared habitats. The manuscript is generally well presented, with a clear introduction, hypothesis, and discussion. However, several major issues should be addressed before it can be considered further:
Major concern:
- Abstract: The authors described three primary mechanisms, including (i) behavioral modulation of neighboring plants, (ii) induction of allelochemical biosynthesis, and (iii) microbial community restructuring, yet propose three different key functional aspects consisting of (i) root system architectural modifications, (ii) defense priming activation, and (iii) coordinated phenotypic plasticity. This inconsistency lacks logical coherence. The abstract should be revised for alignment and clarity.
- Section 2 (Composition of Root Exudates): The manuscript states that “Table 1 is a new classification system for root exudates.” This is not accurate, as the classification is largely based on conventional chemical principles. Moreover, categorizing compounds by molecular weight or regulatory factors is not appropriate here. As this is a core section, the classification should instead focus on functional groups relevant to the objectives of the review. Missing groups such as volatile compounds, lactones (momilactone in rice is a promising compound), etc., should also be added.
- Section 3: This section is too lengthy and should be made more concise.
- Sections 4.1 and 5.2: The mechanisms by which specific root exudate compounds may serve as indicators for kin recognition are only described in very general terms. These mechanisms should be discussed in greater detail.
- Methodology: The strategy for literature search, inclusion/exclusion criteria, and visualization tools used in the figures need to be clearly described.
Minor
- Excessive numbering of listed points makes the text less reader-friendly. Numbering should be used selectively, only when emphasizing information critical to the study objectives.
- Plant species names must be italicized consistently.
- Compound names and chemical formulas should be standardized.
- The text shows signs of overreliance on AI-generated outputs, such as frequent use of em dashes (—) and improper subscripting of chemical formulas. The authors should carefully revise the manuscript to remove these inconsistencies.
I recommend a major revision. The manuscript requires substantial reorganization and refinement to improve logical flow, ensure scientific accuracy, and enhance clarity.
Reviewer 4 Report
Comments and Suggestions for Authors
The author reported in this review article “Advances in Plant Species Recognition Mediated by Root Exu- dates: A Review”. The manuscript offers a comprehensive and insightful review on the role of root exudates as central signaling agents in belowground plant–plant interactions, highlighting their importance in kin recognition, self/non-self-discrimination, and the regulation of physiological and ecological responses in adjacent plants. The authors present a well-organized synthesis of recent literature, effectively capturing the chemical diversity of root exudates, the environmental factors governing their secretion, and the intricate signaling pathways through which they mediate interplant communication. Particularly valuable is the clear mechanistic framework describing how root exudates influence plant fitness through modifications in root system architecture, activation of defense priming, and coordination of phenotypic plasticity, thereby linking molecular events to community-level ecological outcomes. This perspective advances the understanding of root exudates beyond their traditional role as metabolic by-products, positioning them as active mediators of plant behavior and ecosystem functioning. While the review is timely and thorough, it could be further strengthened by a more detailed discussion on cutting-edge metabolomics and imaging techniques for exudate profiling, as well as a deeper exploration of how these findings may be translated into applied contexts, such as crop improvement, soil health management, and sustainable agriculture.
Overall, this review represents a valuable contribution to the field of rhizosphere ecology and will be of significant interest to researchers investigating plant signaling, soil microbial dynamics, and ecosystem interactions. I believe it will be of great interest to the readers of Plants (MDPI Journals). I will make some suggestions that may improve the reach of the paper (in terms of reaching a broad audience). As the modifications can be addressed in straight forward manner and this study represents a significant contribution to the field of rhizosphere ecology and will be of significant interest to researchers investigating plant signaling, soil microbial dynamics, and ecosystem interactions. I am recommending for the publication of this article after the following recommended changes in the manuscript.
- Could the authors clarify how they prioritized the selection of studies discussed in this review? Were specific criteria (e.g., year, methodology, ecological relevance) used to include or exclude literature?
- While the manuscript outlines the broad classes of root exudates, could the authors provide a more detailed comparison of which chemical groups (e.g., flavonoids, phenolics, terpenoids) are most strongly associated with kin recognition versus allelopathic interactions?
- The review highlights exudate-mediated root system architectural changes and defense priming. Are there specific signaling pathways or molecular receptors that have been experimentally validated in these processes, and if so, could the authors expand on these?
- How do abiotic stresses such as drought, nutrient limitation, or soil salinity alter the quantity and composition of root exudates, and what implications does this have for interplant signaling under climate change scenarios?
- The manuscript briefly mentions microbial restructuring. Could the authors elaborate on whether microbial communities act primarily as amplifiers, modulators, or filters of exudate-mediated signaling?
- What are the most promising strategies to translate the knowledge of root exudate-mediated interactions into applied agricultural practices—such as breeding for exudate traits, bioengineering, or microbiome management?
- Could the authors propose a conceptual framework or model that integrates root exudate signaling at molecular, physiological, and ecological scales, to guide future experimental designs?
